# LEVERAGING LARGE LANGUAGE MODELS FOR MULTIPLE CHOICE QUESTION ANSWERING

**Joshua Robinson**\* **& David Wingate**
Department of Computer Science
Brigham Young University
`joshua_robinson@byu.edu, wingated@cs.byu.edu`

## ABSTRACT

While large language models (LLMs) like GPT-3 have achieved impressive results on multiple choice question answering (MCQA) tasks in the zero, one, and few-shot settings, they generally lag behind the MCQA state of the art (SOTA). MCQA tasks have traditionally been presented to LLMs like cloze tasks. An LLM is conditioned on a question (without the associated answer options) and its chosen option is the one assigned the highest probability after normalization (for length, etc.). A more natural prompting approach is to present the question and answer options to the LLM jointly and have it output the symbol (e.g., "A") associated with its chosen answer option. This approach allows the model to explicitly compare answer options, reduces computational costs, and mitigates the effects of tokenization scheme and answer option representations on answer selection. For the natural approach to be effective, the LLM it is used with must be able to associate answer options with the symbols that represent them. The LLM needs what we term multiple choice symbol binding (MCSB) ability. This ability varies greatly by model. We show that a model with high MCSB ability performs much better with the natural approach than with the traditional approach across 20 diverse datasets and largely closes the gap with the SOTA, suggesting that the MCQA ability of LLMs has been previously underestimated.

## 1 INTRODUCTION

Current state of the art (SOTA) methods on many multiple choice question answering (MCQA) tasks involve specialized models, extensive per-task engineering, and individualized tuning in general. What if one model could do just as well as each of these models does individually?

This is part of a general vision for so-called foundation models (Bommasani et al., 2021). Foundation models include large pre-trained language models (LLMs) that have derived enough broad knowledge (spanning, for example, linguistic, factual, and commonsense (Liu et al., 2019; Amrami & Goldberg, 2018; Petroni et al., 2020; Bosselut et al.; Bouraoui et al.; Zuo et al., 2018; Bhagavatula et al., 2019)) to transfer from a simple language modelling objective to a huge array of natural language tasks.

Interestingly, while LLMs have achieved SOTA results on many tasks, they generally fall short on MCQA. Why is this the case, given their general language modelling prowess as suggested by the low cross-entropy loss they attain with all their parameters, data, and compute (Kaplan et al., 2020; Henighan et al., 2020; Hernandez et al., 2021)? Should they not excel, or at least be highly competitive?

In this paper, we argue that they fall short because dominant methods used with them conflate *probabilities of sentences* with *probabilities of correct answers*. We hypothesize that there are fundamental problems with the near-universal approach to MCQA for LLMs, which we refer to as "cloze prompting" (CP). Specifically, these problems include 1) the conflation of the grammaticality, commonality, and "naturalness" of a text and its likelihood qua question-answer, 2) the computational expense of scoring multiple candidate answers, 3) the fact that the LLM cannot explicitly reason

---

\*Work done while at Brigham Young University. Now at University of Southern California.

about and compare different candidate answers, and 4) finicky normalization due to tokenization schemes. The centerpiece of our paper is an extensive investigation of an alternative: we explain how these problems might be solved by what we call *multiple choice prompting* (MCP). In MCP, the language model receives both the question and also a list of candidate answers as on a multiple choice test, with each answer associated with (or "bound" to) a symbol such as "A", "B", "C", etc. We explain how this approach might be why MCP outperforms CP in Section 3.

More importantly, though, we demonstrate that when we prompt LLMs with MCP instead of CP, performance often dramatically improves – approaching or even surpassing SOTA performance. On a varied group of 20 datasets, we show that MCP outperforms CP on all but 4 of the datasets, with a mean gap of 9.7% on all tasks and a max gap of 44%. MCP surpasses old SOTA scores on 9 of 20 datasets (by as much as 15% on a single task), and averaged across all datasets, MCP scores fall 0.6% shy of SOTA.

This implies that the de facto method for prompting LLMs has led them to be considerably under-estimated for MCQA, and that there exists a better general way to prompt a single LLM that scores within a percent of accuracy of all other previous SOTA scores, on average. For the 20 different datasets we consider, SOTA accuracy required 14 customized models and approaches – nearly three individualized setups for every four datasets. We argue that the fact that MCP is comparable to or surpasses SOTA, with no task-specific tuning, is evidence for the efficiency, generality, and overall promise of foundation models in MCQA.

Our primary contribution is three-fold: 1) We present an argument for multiple-choice prompting over cloze prompting and formally define multiple choice symbol binding (MCSB), a required ability for an LLM to benefit from MCP; 2) We show that not all LLMs are equally skilled in this regard; and 3) Across 20 diverse datasets, we show that the models most capable of MCSB can individually approach or beat SOTA on most of the considered tasks when prompted with multiple choice prompting instead of the near-universal approach of cloze prompting. Code is available.[1]

## 2 RELATED WORK

Transformers (Vaswani et al., 2017) have revolutionized the field of NLP by allowing models to effectively absorb much larger datasets via massive scaling in parameter count and compute; these three factors are proportional to lower loss in models (Kaplan et al., 2020; Henighan et al., 2020; Hernandez et al., 2021). Parameter counts have quickly grown from 1.5B in 2018 (Radford et al., 2018) to 540B in 2022 (Chowdhery et al., 2022), and in general, larger models are tested on a more extensive suite of tasks to test their capacity for transfer. This invariably includes multiple choice question answering tasks, and nearly every LLM we know of uses cloze prompting for these tasks (Brown et al., 2020; Du et al., 2022; Smith et al., 2022; Chowdhery et al., 2022; Lieber et al., 2021).

It was, in part, these massive language models that prompted the coining of the phrase "foundation models" (Bommasani et al., 2021). This is a family of large models that are heavily trained on enormous datasets in a self-supervised fashion. They derive general knowledge about a modality and can transfer with impressive sample efficiency to a great number of downstream tasks. A key part of the vision of these models is that they can be repurposed, avoiding the energy, storage, and human capital costs associated with ad hoc models. Our work supports this vision of LLMs as one such foundation model by demonstrating their ability to answer many kinds of multiple choice questions correctly in a zero or few-shot fashion when prompted appropriately.

To the best of our knowledge, the only LLM papers that use the MCP approach for evaluation on any dataset are Gopher (Rae et al., 2021) and followup Chinchilla (Hoffmann et al., 2022). The use of MCP in these works is peripheral and limited to a few specific datasets (MMLU ((Hendrycks et al., 2021)), RACE (Lai et al., 2017), TruthfulQA (Lin et al., 2021b)). One other recent work (Liévin et al., 2022) used MCP when evaluating InstructGPT (Ouyang et al., 2022) on three medical question datasets. In these works the impact on results of the MCP approach in particular is not explored. Ours is the first work to systematically investigate the benefits of this prompting strategy. We show that language models vary greatly in their ability to leverage MCP, and demonstrate that MCP can substantially improve LLM accuracy across a diverse set of tasks. We hope this observation will lead to wider adoption of MCP in LLM work.

---

[1]https://github.com/BYU-PCCL/leveraging-llms-for-mcqa

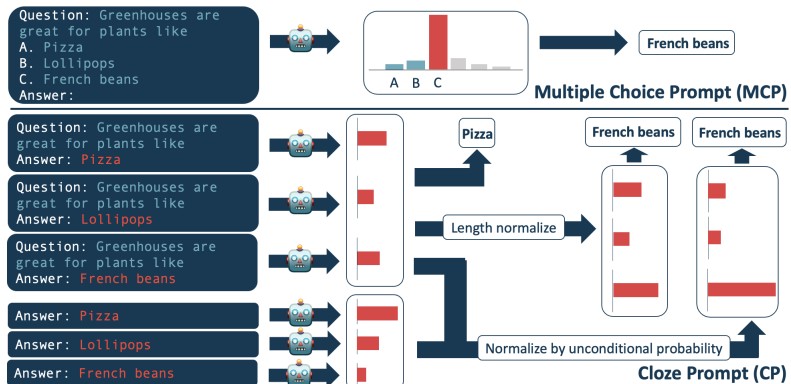

Figure 1: Visualization of the multiple choice prompt (MCP) and cloze prompt (CP) for an example question. Given the same raw data (question and candidate answers) the answer selected when using CP depends on choice of normalization strategy. Question taken from OpenBookQA (Mihaylov et al., 2018) with slight modification.

We are not the first to notice prompting's impact on LLM performance. A whole subfield of prompt engineering exists, with papers suggesting methods for ranking prompts based on mutual information (Sorensen et al., 2022), showing that LLMs suffer from majority label and recency biases (Zhao et al., 2021), and arguing that most few-shot learning is not, in fact, few-shot (Perez et al., 2021).

Given that CP has, up to this point, been the de facto prompting method for LLMs on MCQA tasks, several works have endeavored to improve it with normalization schemes that outperform those used in Brown et al. (2020). These include Contextual Calibration (Zhao et al., 2021) and Domain Conditional PMI (Holtzman et al., 2021). In a similar vein, Izacard et al. (2022) use MCP for evaluation of Atlas on MMLU (Hendrycks et al., 2021) and increase answer ordering invariance by running one forward pass for each cyclic permutation of answer choices then adding model output distributions for each permutation together for use in determining the model's chosen answer.

While in this work we consider methods for effectively leveraging LLMs for MCQA tasks, past work has successfully used many types of models for these tasks. A wide selection of these models are referenced in Table 2. Two notable examples of models that perform well across question answering datasets are UnifiedQA (Khashabi et al., 2020) and UNICORN Lourie et al. (2021).

## 3   CLOZE VS. MULTIPLE CHOICE PROMPTING

In this section, we specify what is meant by cloze prompting and multiple choice prompting, and enumerate the problems that are inherent to the former and ameliorated by the latter.

In cloze prompting, a question is passed to an LLM, the candidate answers are each independently scored by the model, and the answer option selected by the model is chosen to be the one assigned the highest probability. Recognizing that probabilities of answers might be skewed by especially common or uncommon tokens or sequences of varying length, Brown et al. (2020) made use of two different normalization procedures. In one, the sequence's probability is normalized for length by taking the $n$th root, or $P(x_1, x_2, ..., x_n) = \sqrt[n]{\prod_{i=1}^{n} P(x_i)}$. In the other, the answer's probability is normalized by the unconditional probability of the answer, or $\frac{P(\text{completion}|\text{context})}{P(\text{completion}|\text{answer\_context})}$ where answer_context is the string "Answer:     ". In the remainder of the the paper, we refer to length normalization as LN, unconditional normalization as UN, and neither as Raw. See Figure 1 for a visualization of how these strategies work.

In MCP, on the other hand, a question and its symbol-enumerated candidate answers are all passed to an LLM as a single prompt. The prompt is structured such that the LLM must only predict a single token (such as "A", "B", etc.). The answer choice associated with the token assigned the highest probability by the model is chosen to be the model's answer. The probabilities of these

symbols therefore serve as a proxy for each answer's probability. There are several problems with cloze prompting that do not apply to multiple choice prompting:

**Conflation of likelihood as answer and likelihood as natural language** One outstanding problem with CP is that the likelihood of the answer's text could be conflated with the likelihood of the text as an answer. For example, if asked which quote is from Shakespeare's *Macbeth*, the phrase `Give sorrow words; the grief that does not speak knits up o-er wrought heart and bids it break.` might be less common or grammatical than other sentences, which could artificially deflate its score. MCP does not face this problem because there is no grammar to alter the score.

**Reliance on normalization procedures** With CP, use of special normalization strategies are typically essential for achieving high performance. These often incur a computational cost or depend on choice of tokenization scheme. With MCP, there is no need for any normalization strategy.

**No direct comparison between answers** In CP, candidate answers are not compared to each other except implicitly through their final probabilistic scores. MCP gives LLMs the ability to explicitly compare and contrast different answer options. This makes LLM+MCP more comparable to SOTA methods that typically have all answer options presented at once. Additionally, provision of answer choices to LMs is essential for response calibration (Kadavath et al., 2022).

**Expense** Lastly, cloze prompting is computationally expensive. For a question with $n$ possible answer choices, CP requires $n$ forward passes through the LLM for the Raw or LN normalization strategies, and $2n$ forward passes for the UN strategy. MCP only requires a single pass (itself slightly cheaper than CP forward passes because the model only needs to generate a single output token).

## 4 THE CHALLENGE OF MULTIPLE CHOICE SYMBOL BINDING

When presenting a multiple choice question, the candidate answers must be enumerated in some order. Humans' answers to such questions are generally order-invariant. If an LLM exhibits the same characteristic, we say that it is capable of *multiple choice symbol binding* (MCSB). Interestingly, LLMs vary substantially in terms of this ability.

Consider the multiple choice prompt example from Figure 1. Given the answer order "Pizza", "Lollipop", "French beans" (as shown in the figure) GPT-3 (Davinci) Brown et al. (2020) assigns the highest probability to the token "A," which is associated with pizza. However, if we change the ordering to "French beans", "Lollipops", "Pizza", GPT-3 surprisingly still assigns the highest probability to "A," which is now associated with French beans. Simply changing the order of the candidate answers changes the model's answer.

How can we compare the relative symbol binding ability of two models? One way is to measure what we term *Proportion of Plurality Agreement* (PPA). Given a question with $n$ answer options, there are $n!$ different ways these options can be associated with an ordered, fixed set of symbols. To measure PPA for a given model and question we present that question to the model with each different ordering and for each ordering record the answer assigned the highest probability by the model. PPA for that question is the proportion of orderings that chose the plurality answer among all orderings. For a dataset, PPA is averaged over all questions. Importantly, PPA measures order invariance irrespective of model ability to perform a task. If a model performs poorly on a task but answers consistently across possible orders of answer options it will still have a high PPA. For a dataset where each question has $n$ answer choices, the baseline PPA is $1/n$.

Using PPA, we compare several popular LLMs' MCSB ability. The first is GPT-3 (Brown et al., 2020) (Davinci). We also consider Codex (Chen et al., 2021) (Davinci) and InstructGPT (Ouyang et al., 2022) (Curie and Davinci). These two models were fine-tuned from GPT-3's weights for code modelling and instruction following, respectively. We also evaluate the MCSB ability of GPT-2 Radford et al. (2019), CodeParrot (Tunstall et al., 2022) (GPT-2 fine-tuned on code), and Jurassic-1 (Lieber et al., 2021) (Jumbo). The parameter count of these models spans from 1.5B (GPT-2) to 178B (Jurassic-1 Jumbo). We use API requests for GPT-3, Codex, Instruct, and Jurassic-1. For GPT-2 and CodeParrot we use checkpoints from Hugging Face Transformers (Wolf et al., 2020).

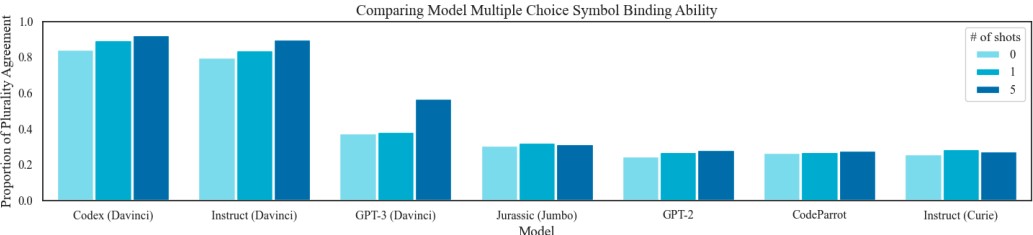

Figure 2: Comparison of the multiple choice symbol binding ability of different models as measured by PPA on a subset of OpenBookQA (Mihaylov et al., 2018).

We evaluate on the OpenBookQA dataset (Mihaylov et al., 2018), a multiple choice QA dataset composed of 500 science questions. We randomly sample 100 instances from the dataset to reduce computational cost. Each question has four answer options, meaning that the PPA random baseline is 25%. We choose to use OpenBookQA because of its relatively small size, because it has been used widely for benchmarking LLMs, and because it is an archetypal multiple choice dataset with questions modeled after human multiple choice exams. As in other work with LLMs, we do not explicitly make the "book" for the dataset (a list of elementary science facts) available to the models.

The results of our evaluation can be found in Figure 2. The most immediately interesting result is that Codex (Davinci) and Instruct (Davinci) significantly outperform the other models in terms of PPA, showing remarkable invariance to answer ordering. GPT-3 seems to perform about half as well, interestingly outperforming the larger Jurassic-1 (Jumbo) model. GPT-2, CodeParrot, and Instruct (Curie) all have PPA scores close to the 25% baseline.

Another apparent trend is that providing exemplars increases PPA consistently across models. This is especially notable for GPT-3. While we do not explore what about these models causes high MCSB ability, it appears that model size could be an important factor because Instruct Davinci has a much higher PPA than Instruct Curie. This hypothesis is in line with the work of Wei et al. (2022).

It also seems like further training (on source code or with reinforcement learning based on human preferences) is essential (Codex and Instruct both outperform GPT-3 substantially even though they are fine-tuned versions of it). It makes sense that training on code increases symbol binding ability because performing well on a next token prediction task for code requires grasping symbol binding as it is used in e.g., dictionaries and list indexing. Training on code alone does not seem sufficient in and of itself, though, because CodeParrot did not achieve notably better PPA than did GPT-2.

Given the strong multiple choice symbol binding ability of Codex and Instruct, a natural next question is whether using these models with MCPs results in higher accuracy. We address this question in the next section.

## 5 EXPERIMENTAL SETUP

We evaluate the performance of a model with strong MCSB ability and multiple choice prompts across a set of 20 diverse datasets. In this section we discuss how that model was chosen (Section 5.1) and which datasets we use for evaluation (Section 5.2). We also address the possibility of dataset leakage into model training data (Section 5.3) and explain prompt engineering choices (Section 5.4).

### 5.1 MODELS

In Section 4 we observed that models like Codex (Davinci) (Chen et al., 2021) and Instruct (Davinci) (Ouyang et al., 2022) demonstrated much stronger MCSB ability than did GPT-3. In this section we explore whether higher MCSB ability leads to higher multiple choice task accuracy. We evaluate 5-shot model performance on a commonsense reasoning dataset (OpenBookQA (Mihaylov et al., 2018)), a cloze/completion dataset (StoryCloze (Mostafazadeh et al., 2016)), and a reading comprehension dataset (RACE-m (Lai et al., 2017)). See Section 5.2 for more information on these datasets. We randomly sample 500 instances for both StoryCloze and RACE-m to reduce computational costs.

| Dataset | GPT-3 | | | | Instruct | | | | Codex | | | |
|---------|------|------|------|------|------|------|------|------|------|------|------|------|
|         | Raw | LN | UN | MCP | Raw | LN | UN | MCP | Raw | LN | UN | MCP |
| OpenBookQA | 35.0 | 46.8 | **57.4** | 41.4 | 41.8 | 49.6 | 58.4 | **77.4** | 43.0 | 51.4 | 65.6 | **83.0** |
| StoryCloze | 75.2 | **76.4** | 75.6 | 70.8 | 78.0 | 78.8 | 82.4 | **97.6** | 80.8 | 83.6 | 84.0 | **97.4** |
| RACE-m | 55.6 | **57.2** | 56.6 | 50.2 | 63.2 | 64.8 | 66.8 | **89.6** | 63.4 | 67.0 | 63.8 | **89.2** |

Table 1: Comparison of large language model performance across prompting strategies. The three cloze prompting normalization strategies are described in Section 3. MCP is multiple choice prompting. The best accuracy for each model and dataset is bolded.

The results of our comparison can be found in Table 1. Whereas choosing answer options based on cloze prompts (with max raw probability (Raw), max raw probability after length normalization (LN), or max raw probability after unconditional normalization (UN)) performs best for GPT-3, MCP always performs best for Instruct and Codex, and it does so by a sizeable margin. Because these models have high MCSB ability they are able to effectively leverage MCP prompts to achieve higher accuracy. Instruct and Codex both outperform GPT-3 by large margins across all tasks.

It is evident that both Codex and Instruct have high multiple choice symbol binding ability (see Figure 2) and can effectively leverage MCP prompts across tasks (see Table 1). We make no argument that one is empirically stronger than the other. For all our further experiments we choose to use Codex (Davinci) because it is the least expensive (since it is currently in free beta), and because Codex should not add any dataset leakage issues that were not already present in GPT-3 since it was exclusively fine-tuned on Python files.

## 5.2 DATASETS

We compare multiple choice prompts and cloze prompts across a diverse array of datasets. Examples of questions from each of these datasets can be found in Appendix A.

**Common sense reasoning** ARC (Clark et al., 2018) consists of grade-school science questions. Its challenge set is filtered down to questions not correctly answered by simple retrieval and co-occurence solvers. CODAH (Chen et al., 2019) questions were adversarially generated to fool BERT-base Devlin et al. (2019) fine-tuned on SWAG (Zellers et al., 2018). CommonsenseQA (Talmor et al., 2019) questions are based on ConceptNet (Speer et al., 2017) concepts, allowing for diversity and incorporation of world knowledge into questions. COPA (Roemmele et al., 2011) is a causal reasoning dataset that tasks models with determining the cause or effect of a premise. Composed of questions about metaphors, Fig-QA (Liu et al., 2022a) is designed to test nonliteral reasoning abilities. MedMCQA (Pal et al., 2022) has questions from many areas of the medical domain drawn from medical exams (see Appendix F for a full list of subjects covered). The Massive Multitask Language Understanding (MMLU) benchmark (Hendrycks et al., 2021) consists of diverse tasks from STEM, the humanities, and the social sciences (a full list of tasks can be found in Appendix G). OpenBookQA (Mihaylov et al., 2018) endeavors to test world knowledge and reasoning using science questions. PIQA (Bisk et al., 2020) questions focus on the area of commonsense reasoning related to understanding of physical interactions with the world. RiddleSense (Lin et al., 2021a) and Social IQa (Sap et al., 2019) are designed to test for model ability to reason about riddles and social situations respectively.

**Natural language inference** ANLI (Nie et al., 2020) is a dataset of adversarially generated NLI questions. These questions were generated by annotators in three rounds, with annotators seeking to generate questions challenging for increasingly capable models trained for NLI.

**Cloze and completion tasks** HellaSwag (Zellers et al., 2019) tasks models with predicting the best continuation for a video caption or WikiHow article. StoryCloze (Mostafazadeh et al., 2016) is also a continuation prediction task, but with short, four-sentence stories.

**Text classification** AG News (Zhang et al., 2015) is a news classification task.

**Winograd-style tasks** Winogrande (Sakaguchi et al., 2021) is a set of winograd schema questions that was carefully crowdsourced and then filtered with a bias mitigation algorithm. Because we are doing evaluation without fine-tuning in this work we evaluate in the smallest training data setting (XS), but also include results in the largest setting (XL) to facilitate comparison with prior work.

**Reading comprehension** Cosmos QA (Huang et al., 2019) is a commonsense reasoning based reading comprehension task designed to test model ability to "read between the lines." The questions in the DREAM (Sun et al., 2019) dataset are based on dialogues. LogiQA (Liu et al., 2020) tasks models with answering questions from a human exam that tests critical thinking. RACE (Lai et al., 2017) is a widely used reading comprehension dataset with questions taken from middle and high school English exams for Chinese students.

## 5.3    ADDRESSING DATASET LEAKAGE

A concern in all work with LLMs is that an LLM being used for evaluation may have been exposed to the contents of an evaluation dataset during pre-training. We only evaluate on Codex, which was initialized from GPT-3's weights and fine-tuned exclusively on Python files. Thus the risks of dataset leakage we face are effectively the same as those faced by Brown et al. (2020).

The dataset leakage risk of the 9 of our datasets GPT-3 was tested on in the GPT-3 paper was evaluated extensively by the authors of that paper. Although we were not able to acquire the GPT-3 training data to perform a manual collision check for the other datasets, there are several reasons why we suspect dataset leakage is not meaningfully impacting our results. First, we note (anectdotally) that Codex+MCP errors in evaluation occur disproportionately on questions that are ill-defined (not solvable by humans, having multiple answers, etc.) (see non-cherry-picked examples in Appendix D). If there were training set leakage, this set would be more independent of being well-formed than it is; that is, failures by the model seem to be due more to the deck being stacked against good reasoning than to the model having failed to memorize the data. Second, shuffling symbol-enumerated candidate answer ordering does not systematically harm performance (see Appendix C). Third, CP+MCP performance on public and private test sets is generally comparable.

The strongest argument that MCP's performance is not determined by training set leakage is CP's meaningfully inferior performance for the same model; if the model had memorized the training data, it would have ascribed more probability to the correct answer regardless of prompting method.

## 5.4    PROMPT PHRASING

In our experiments we try to keep the comparison of cloze prompts and multiple choice prompts as simple and fair as possible. Unlike Brown et al. (2020) we do not tune $K$, the number of few-shot exemplars, based on a development set, nor do we develop highly task-specific prompt phrasings. $K$ is always chosen to be as high as possible while respecting Codex's 4,000 token context limit.

Our prompt phrasing is consistent across tasks: We prefix the raw question with `Question:`, list answer options with associated letters (like `A. Lollipop`), and finish prompts with `Answer:`. We measure model probability for an answer via the probability of the symbol associated with it. When a passage is included as part of a question (as in reading comprehension) we insert the passage before the question and prefix it with `Passage:` (`Story:` for StoryCloze (Mostafazadeh et al., 2016) and `Dialogue:` for DREAM (Sun et al., 2019)). See Appendix A for examples. Our prompts are simple and modelled after those in Brown et al. (2020).

Our goal in this work is to provide a fair comparison between cloze prompts and multiple choice prompts, and not to maximize accuracy by extensive prompt engineering. However, we can say anecdotally that multiple choice prompts seem very robust to different wordings and symbol choices. Further prompt engineering for MCPs could be interesting future work.

## 6    RESULTS

The results of our experiments can be found in Table 2. In the table the CP columns represent cloze prompting with the best possible strategy on the test set. That is, for each dataset and cloze prompts we calculated test accuracy based on raw accuracy, raw accuracy with length normalization, and

| Dataset | N | K | Zero-Shot | | One-Shot | | Few-Shot | | Server | SOTA |
|---|---|---|---|---|---|---|---|---|---|---|
| | | | CP | MCP | CP | MCP | CP | MCP | | |
| AG News | 4 | 38 | 68.2 | **83.5** | 77.6 | **87.1** | **90.1** | 89.4 | | 95.6[a] |
| ANLI R1 | 3 | 27 | **45.3** | 33.2 | 35.6 | **61.7** | 58.4 | **64.2** | | 75.5[b] |
| ANLI R2 | 3 | 26 | **39.2** | 33.6 | 35.7 | **53.0** | 51.8 | **55.2** | | 58.6[c] |
| ANLI R3 | 3 | 26 | **37.8** | 34.3 | 35.5 | **47.8** | 54.2 | **54.5** | | 53.4[c] |
| ARC (Challenge) | 4 | 50 | 58.9 | **81.7** | 64.1 | **82.8** | 66.6 | **86.1** | | 86.5[d] |
| ARC (Easy) | 4 | 57 | 84.2 | **93.1** | 85.9 | **93.5** | 87.8 | **94.7** | | 94.8[d] |
| CODAH | 4 | 63 | 56.8 | **76.0** | 65.4 | **87.8** | 73.6 | **91.9** | | 84.3[e] |
| CommonsenseQA | 5 | 79 | 68.5 | **72.0** | 73.1 | **78.9** | 78.6 | **83.2** | 76.6 | 79.1[f] |
| COPA | 2 | 113 | **92.0** | 89.0 | 95.0 | **99.0** | 96.0 | **100.0** | — | 99.2[d] |
| Cosmos QA | 4 | 24 | 43.0 | **75.5** | 44.0 | **81.8** | 38.1 | **82.4** | 83.5 | 91.8[g] |
| DREAM | 3 | 7 | 72.7 | **91.3** | 82.5 | **93.3** | 84.3 | **94.1** | | 92.6[h] |
| Fig-QA | 2 | 99 | 79.6 | **84.7** | 82.4 | **86.7** | 82.5 | **94.0** | 93.1 | 90.3[i] |
| HellaSwag | 4 | 16 | — | 71.0 | — | 75.1 | — | 73.6 | — | 93.9[g] |
| LogiQA | 4 | 16 | 36.6 | **44.5** | 37.5 | **45.3** | 37.8 | **47.3** | | 42.5[j] |
| MedMCQA | 4 | 58 | 37.8 | **52.1** | 42.1 | **53.9** | 41.2 | **54.4** | 58.0 | 41.0[k] |
| MMLU | 4 | 5 | 49.5 | **62.1** | — | 68.2 | — | 69.5 | | 67.5[l] |
| OpenBookQA | 4 | 83 | 63.2 | **72.0** | 64.0 | **81.6** | 71.2 | **87.0** | | 87.2[f] |
| PIQA | 2 | 35 | **83.7** | 73.7 | **84.1** | 81.8 | **86.1** | 84.5 | — | 90.1[g] |
| RACE-h | 4 | 4 | 52.3 | **82.1** | 53.2 | **85.1** | 55.2 | **86.2** | | 89.8[m] |
| RACE-m | 4 | 8 | 67.5 | **85.4** | 70.5 | **89.3** | 71.7 | **90.3** | | 92.8[m] |
| RiddleSense | 5 | 59 | **79.8** | 67.6 | **89.1** | 77.1 | **91.3** | 83.9 | 80.0 | 68.8[f] |
| Social IQa | 3 | 72 | 52.1 | **64.4** | 58.1 | **72.2** | 62.4 | **74.9** | 76.0 | 83.2[g] |
| StoryCloze | 2 | 44 | 80.3 | **97.5** | 83.4 | **98.3** | 88.2 | **98.5** | | 89.0[n] |
| Winogrande (XL) | 2 | 102 | 62.5 | **64.5** | 71.6 | **71.6** | 75.5 | 72.1 | 72.3 | 91.3[g] |
| Winogrande (XS) | 2 | 102 | 63.0 | **64.8** | 71.0 | **71.3** | 76.2 | 73.6 | 73.8 | 79.2[g] |

Table 2: Comparison of multiple choice prompt (MCP) and cloze prompt (CP) with Codex. N is the number of answer options for each question. K exemplars are provided in the few-shot setting. SOTA values come from [a]Yang et al. (2019), [b]Wang et al. (2021), [c]Lan et al. (2019), [d]Zoph et al. (2022), [e]Yang et al. (2020), [f]Khashabi et al. (2020), [g]Lourie et al. (2021), [h]Zhang & Yamana (2022), [i]Liu et al. (2022b), [j]Jiao et al. (2022), [k]Gu et al. (2020), [l]Hoffmann et al. (2022), [m]Jiang et al. (2020), and [n]Chowdhery et al. (2022). These values are computed using a private test set when it exists (for rows with a Server value) or on a public test set otherwise. The best prompt method for each dataset and exemplar count is bolded. The SOTA including our experimental results is underlined. Values marked with — could not be computed - mostly due to computational restraints (see Appendix B).

raw accuracy with unconditional normalization. The CP column contains *the highest accuracy of any strategy*. This is an unrealistically strong baseline, since no single normalization scheme is universally optimal. The results of each individual scheme on all datasets can be found in Table 5.

The results in Table 2 validate our hypothesis that for a model with high MCSB ability (Codex) using MCP outperforms CP. This holds consistently across datasets and number of exemplars. MCP increases accuracy over CP by 8.3, 12.2, and 9.7 percentage points on average in the zero, one, and few-shot settings, respectively. This improved performance also comes without reliance on specialized normalization procedures, and with 4.3x less API calls (or forward passes of batch size 1) than the chosen CP strategies across tasks and exemplar settings.

The dataset with the largest gain between the cloze prompts and multiple choice prompts is Cosmos QA. For this dataset, using MCP instead of CP increases accuracy by 32.5, 37.8, and 44.3 pecentage points in the zero, one, and few-shot settings, respectively. This substantial improvement on task performance is likely due to the fact that Cosmos QA questions (including their answer options) have somewhat irregular spacing[2]. This poses no issue for MCPs, but is a serious issue for CPs that rely on the linguistic representation of answer options.

---

[2]https://huggingface.co/datasets/cosmos_qa

| Corruption | OpenBookQA | | | | StoryCloze | | | | RACE-m | | | |
|---|---|---|---|---|---|---|---|---|---|---|---|---|
| | Raw | LN | UN | MCP | Raw | LN | UN | MCP | Raw | LN | UN | MCP |
| None | 43.0 | 51.4 | 65.2 | **82.4** | 81.0 | 83.6 | 83.8 | **97.4** | 63.2 | 66.4 | 64.0 | **89.4** |
| Caps | 31.4 | 43.0 | 49.6 | **79.8** | 63.6 | 71.4 | 70.4 | **96.8** | 50.6 | 57.0 | 52.6 | **88.8** |
| Space | 32.2 | 43.4 | 44.4 | **80.6** | 71.6 | 78.2 | 71.2 | **98.0** | 53.0 | 63.2 | 51.2 | **89.0** |

Table 3: Comparison of Codex accuracy under different answer choice corruptions. The three cloze prompting normalization strategies are described in Section 3. MCP is multiple choice prompting. The best accuracy for each dataset and corruption type is bolded.

To further explore the extent to which MCP benefits from direct comparison between answer choices and from separating likelihood of answer choices and their likelihoods in terms of natural language, we evaluate the 3-shot performance of Codex on the dataset splits used in Section 5.1 under two corruptions of answer choices. For the "Caps" corruption we randomly uppercase or lowercase each character in each answer choice. For the "Space" corruption we randomly add a space before, after, or within each word with at least three characters in each answer choice. Results can be seen in Table 3. Whereas performance for SC across strategies and datasets drops by 12.4% and 10.3% for the "Caps" and "Space" corruptions respectively, these drops are only 1.3% and 0.5% for MCP.

There are four datasets in Table 2 where CP outperforms MCP - AG News, PIQA, RiddleSense, and Winogrande. One thing in common between AG News, Winogrande, and RiddleSense is they tend to have short, often one word answers. For these datasets CP is acting more like MCP because answer option length and wording have less impact. Some PIQA questions have answer options much longer than normal ones, perhaps making MCSB more challenging. Additionally, PIQA questions sometimes appear very "cloze-friendly" (see example prompt in Appendix A).

In addition to consistently outperforming Codex+CP, Codex+MCP sets a new state of the art for 9 datasets. For MedMCQA Codex+MCP has an accuracy 13.4% above the old SOTA model, Pubmed-BERT (Gu et al., 2020). This seems to suggest that, in contrast to prior work (Moradi et al., 2021), large language models may have high potential in the biomedical domain. The key is prompting them in a way that effectively aligns them to the task.

## 7 CONCLUSION

In this work we have argued for multiple choice prompting over the universally-practiced cloze prompting, formalized the idea of multiple choice symbol binding (MCSB), showed that large language models vary greatly in their MCSB ability, and demonstrated that for a model with high MCSB ability like OpenAI Codex (Chen et al., 2021) multiple choice prompts generally elicit much more accurate responses than do cloze prompts. This approach led to a new state-of-the-art on 9 popular datasets, and on average scored within a percentage point of previous SOTA, all using a single model and single prompting approach. This demonstrates the power of LLMs as foundation models and is a strong case study in how these models might be used broadly in the future.

The future for symbol-binding is exciting, with potential to teach LLMs new concepts and symbols representing them and have language models fold these new frameworks and ideas into their already-rich world models.

There is also a concrete lesson to be drawn from seeing such drastic improvement in LLM performance with such a simple change to prompting: an intuitive, sensible change to the way we train and use our models can quickly unlock their potential in ways that we have previously been blind to as a field. This should fill us with motivation to seek out such improvements.

Promising directions for future work include further prompt engineering for multiple choice prompts; evaluating prompts on more datasets, tasks, and models; and assessing what factors are responsible for high MCSB ability. Our results suggest that the performance of large language models on multiple choice question answering tasks has been previously underestimated, and we hope that increased adoption of multiple choice prompts will lead to more effective probing and utilization of large language models.

ACKNOWLEDGEMENTS

The authors gratefully acknowledge the support of the National Science Foundation under grant NSF EAGER 2141680. The opinions, findings, and conclusions, or recommendations expressed are those of the author(s) and do not necessarily reflect the views of the National Science Foundation.

REPRODUCIBILITY STATEMENT

Source code for replicating all experiment results can be found at `https://github.com/BYU-PCCL/leveraging-llms-for-mcqa`. Names of all model checkpoints and API endpoints used can be found in `constants.py`. This file also contains the single random seed we use for selection of few-shot exemplars, strong shuffling, dataset downsampling, and random corruptions.

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

## A    PROMPTS USED FOR EACH DATASET

In this section we provide examples of the prompts used for each dataset. Any typos in questions are present in the datasets they are drawn from. The examples are multiple choice prompts (MCPs). Removing the answer options from an example question yields the cloze prompt (CP) we used for that question.

```
Article: At Disney, Mending Fences or Moving On? Without Michael D.
Eisner at the helm of the Walt Disney Company, will Harvey Weinstein and
Steven P. Jobs stay as partners?
Question: What is the best classification for this article?
A. World
B. Sports
C. Business
D. Sci/Tech
Answer:
```

Figure 3: Prompt example for the AG News dataset.

```
Premise: press release: Did you know that Marquette University owns the
original manuscripts for J. R. R. Tolkien's The Hobbit and The Lord of
the Rings? William Fliss, Archivist in Marquette's Department of Special
Collections and University Archives, will share the remarkable tale of
how these literary treasures came to Wisconsin, and he will explain what
these manuscripts can tell us about one of the most iconic authors of the
twentieth century. Cost: Suggested
donation of $3/person
Hypothesis: Attendees will pay $3.
A. Hypothesis is definitely true given premise
B. Hypothesis might be true given premise
C. Hypothesis is definitely not true given premise
Answer:
```

Figure 4: Prompt example for the ANLI dataset. Wording was taken from Gururangan et al. (2018).

```
Question: What adaptation is necessary in intertidal ecosystems but not
in reef ecosystems?
A. the ability to live in salt water
B. the ability to use oxygen in respiration
C. the ability to cope with daily dry periods
D. the ability to blend into the surroundings
Answer:
```

Figure 5: Prompt example for the ARC dataset.

```
Question: Two friends are looking at cakes in a bakery. They
A. start throwing water at each other.
B. pay the bartender for the cake and leave the pub.
C. run a marathon.
D. order a cheesecake.
Answer:
```

Figure 6: Prompt example for the CODAH dataset.

```
Question: How does a bishop move from one place to another?
A. chess game
B. church
C. in a car
D. queen
E. cathedral
Answer:
```

Figure 7: Prompt example for the CommonsenseQA dataset.

```
Question: The pond froze over for the winter so
A. People skated on the pond.
B. People brought boats to the pond.
Answer:
```

Figure 8: Prompt example for the COPA dataset.

```
Passage: Today I went to the new Trader Joe 's on Court Street . It is so
pretty . It 's inside what appears to be an old bank . It was spacious
and there were no NYU students wearing velour sweatpants .
Question: What was the narrator very impressed with ?
A. None of the above choices .
B. The grocery store .
C. The NYU campus .
D. The bank workers .
Answer:
```

Figure 9: Prompt example for the Cosmos QA dataset.

```
Dialogue: M: I want to send this package by first-class mail. W: Do you
want it insured? M: Yes, for 50 dollars, please. I'd also like some
stamps--a book of 22 and three airmail. W: You'll have to get those at
the stamp window over there, next to general delivery. M: Can I get money
orders there, too? W: No, that's to the left, three windows down the hall.
Question: Where can the man get money orders?
A. At the stamp window.
B. Next to general delivery.
C. Three windows down the hall.
Answer:
```

Figure 10: Prompt example for the DREAM dataset.

```
Question: The chihuahua believes it is a wolf, meaning
A. The small dog thinks it is undefeatable
B. The small dog always stays on your lap
Answer:
```

Figure 11: Prompt example for the Fig-QA dataset.

```
Passage: [header] How to get around london easily [title] Know how you're
going to travel. [step] The easiest method of travel in london is the
tube. For this, it is easiest to buy what is called an' oyster card' or a
get a travelcard for all zones from one of the automated machines in a
tube station.
Question: Which choice best continues the passage?
A. People take an oyster card (this is a permanent, digital card) for
optimal services and there are a number of reputable card companies that
buy oyster cards. [title] Firstly, when considering destination, are you
travelling with a package? [step] Do you want to surprise your friends
and family at london.
B. These cover buses, tubes, trams and overground trains throughout
the city. This is usually the best option, especially for tourists,
as you can travel as much as you'd like in one day with one flat fare.
C. [title] Know the locations of the railway stations you are going
to. [step] Look for normal bus lines around london.
D. The card lets you ride on the tube without the added cost of any
rail, bus, or train stops. You can also travel by car (train makes
easier to return for rides in london if you're travelling as
non-railway cars), train from the station, or post office.
Answer:

Passage: (Kayaking) Man is kayaking in a calm river. Man is standing in
te seasore talking to the camera and showing the kayak.
Question: Which choice best continues the passage?
A. man is getting in the sea and sits in a kayak.
B. man is kayaking in rafts and going through mountains.
C. man is kayaking on a snowy river.
D. man is returning in a river with a land trail and a shop.
Answer:
```

Figure 12: Prompt examples for the HellaSwag dataset. We include a WikiHow example (top) and an ActivityNet example (bottom) because they are formatted slightly differently.

```
Question: Statistics show that train accidents in a country mostly occur
in the southern region, so it is safer to travel by train in the northern
region. Which of the following can best refute the above argument?
A. Slower train speeds in the north of the country
B. There are many more train lines in the south of the country than in
the north
C. Many lines in the south of the country already use EMUs
D. Most of the northern part of the country is mountainous and is
more suitable for car driving
Answer:
```

Figure 13: Prompt example for the LogiQA dataset.

```
Question: Keratin in skin is softer than keratin in nail because keratin
in skin has -
A. Less number of disulphide bonds
B. Less number of salt bridges
C. High sodium content
D. Different affinity for water
Answer:
```

Figure 14: Prompt example for the MedMCQA dataset.

```
Question: A state has recently enacted a statute prohibiting the disposal
of any nuclear wastes within the state. This law does not contravene or
conflict with any federal statutes. A man operates a company in the state
that is engaged in the disposal of nuclear wastes. Subsequent to the
passage of the state statute, the man, not yet aware of the new law,
entered into contracts with many out-of-state firms to dispose of their
nuclear wastes in the state. On account of this new law, however, the
man will be unable to perform these contracts. Assume that the man has
standing to challenge this state law. Which of the following presents his
strongest constitutional grounds to challenge the state law prohibitin
the disposal of nuclear wastes within the state?
A. The commerce clause.
B. The equal protection clause of the Fourteenth Amendment.
C. The privileges and immunities clause of Article IV, Section 2.
D. The contract clause.
Answer:
```

Figure 15: Prompt example for the MMLU dataset.

```
Question: Greenhouses are great for plants like
A. Pizza
B. Lollipops
C. Candles
D. French beans
Answer:
```

Figure 16: Prompt example for the OpenBookQA dataset.

```
Question: To clear snot out of your nose,
A. place a tissue over your nose and blow the snot out.
B. place a tissue over your nose and suck the snot in.
Answer:
```

Figure 17: Prompt example for the PIQA dataset.

```
Passage: Food is very important.Everyone needs to eat well if he wants to
have a strong body.Our minds also need a kind of food.This kind of
food is knowledge .
When we are very young,we start getting knowledge.Kids like watching and
listening.Color pictures especially interest them.When kids are older,
they enjoy reading.When something interests them,they love to ask
questions.
Our minds,like our bodies,always need the best food.Studying on our own
brings the most knowledge.
If someone is always telling us answers,we never learn well.When we study
correctly and get knowledge on our own,we learn more and understand
better.
Question: We start getting knowledge   _  .
A. when we are old
B. when we are very young
C. when we are pupils
D. when we are parents
Answer:
```

Figure 18: Prompt example for the RACE dataset.

```
Question: This is an ancient suit that is not worn with a tie
A. shirt
B. armor
C. helmet
D. shirt and trousers
E. hair
Answer:
```

Figure 19: Prompt example for the RiddleSense dataset.

```
Question: Cameron returned home with a bag of candy to eat all night
long. What will Others want to do next?
A. great
B. buy the candy to eat
C. bored
Answer:
```

Figure 20: Prompt example for the Social IQa dataset.

```
Story: Jon loved the night sky. He would spend many of his nights looking
at the stars. His mom saw that he loved the night sky. His mom bought him
a telescope.
Question: Which sentence best completes the story?
A. Jon then watched germs with his microscope.
B. Jon used his telescope often.
Answer:
```

Figure 21: Prompt example for the StoryCloze dataset.

```
Question: So _ plays video games because Leslie has a lot of free time
while Nelson has to work all the time.
A. Leslie
B. Nelson
Answer:
```

Figure 22: Prompt example for the Winogrande dataset.

## B  COMPUTATIONAL CONSTRAINTS

While the OpenAI Codex Beta [3] being free enabled the high volume of experiments we performed, we were limited by its maximum 20 API requests per minute limit (which we werent't able to hit in practice). Computing just the zero-shot CP value for MMLU (Hendrycks et al., 2021) in Table 2 took over a week.

## C  RESULTS UNDER STRONG SHUFFLE OF ANSWER OPTIONS

---

[3] https://openai.com/blog/openai-codex/

| Dataset | N | K | Zero-Shot | | One-Shot | | Few-Shot | |
|---|---|---|---|---|---|---|---|---|
| | | | No | Shuffle | No | Shuffle | No | Shuffle |
| AG News | 4 | 38 | 83.5 | — | 87.1 | — | 89.4 | — |
| ANLI R1 | 3 | 27 | 33.2 | **38.2** | **61.7** | 54.0 | 64.2 | **66.2** |
| ANLI R2 | 3 | 26 | 33.6 | **34.8** | **53.0** | 48.1 | 55.2 | **57.1** |
| ANLI R3 | 3 | 26 | 34.3 | **35.4** | 47.8 | **50.1** | 54.5 | **56.3** |
| ARC (Challenge) | 4 | 50 | 81.7 | **82.2** | 82.8 | **83.0** | **86.1** | 85.6 |
| ARC (Easy) | 4 | 57 | **93.1** | 92.8 | **93.5** | 93.4 | **94.7** | 94.4 |
| CODAH | 4 | 63 | **76.0** | 75.9 | **87.8** | 87.5 | 91.9 | **92.5** |
| CommonsenseQA | 5 | 79 | **72.0** | 71.2 | **78.9** | 78.0 | **83.2** | 82.9 |
| COPA | 2 | 113 | **89.0** | 86.0 | **99.0** | 96.0 | **100.0** | 99.0 |
| Cosmos QA | 4 | 24 | 75.5 | **76.3** | 81.8 | **82.8** | 82.4 | **83.7** |
| DREAM | 3 | 7 | 91.3 | **91.8** | 93.3 | **94.1** | **94.1** | 94.0 |
| Fig-QA | 2 | 99 | **84.7** | 81.8 | **86.7** | 86.0 | **94.0** | 92.5 |
| HellaSwag | 4 | 16 | **71.0** | 70.6 | 75.1 | — | 73.6 | — |
| LogiQA | 4 | 16 | **44.5** | 39.3 | **45.3** | 42.9 | **47.3** | 41.6 |
| MedMCQA | 4 | 58 | **52.1** | 49.4 | **53.9** | 52.5 | **54.4** | 51.9 |
| MMLU | 4 | 5 | 62.1 | — | 68.2 | — | 69.5 | — |
| OpenBookQA | 4 | 83 | 72.0 | **74.8** | 81.6 | **82.0** | **87.0** | **87.0** |
| PIQA | 2 | 35 | 73.7 | **73.9** | **81.8** | 80.5 | 84.5 | **86.2** |
| RACE-h | 4 | 4 | **82.1** | 81.5 | 85.1 | **85.3** | 86.2 | **86.5** |
| RACE-m | 4 | 8 | 85.4 | **86.3** | **89.3** | 89.2 | **90.3** | 90.1 |
| RiddleSense | 5 | 59 | **67.6** | 67.3 | **77.1** | 74.6 | **83.9** | 82.6 |
| Social IQa | 3 | 72 | **64.4** | 64.3 | **72.2** | 71.3 | 74.9 | **75.0** |
| StoryCloze | 2 | 44 | 97.5 | **98.1** | **98.3** | 98.2 | 98.5 | **98.7** |
| Winogrande (XL) | 2 | 102 | **64.5** | 59.5 | **71.6** | 67.8 | **72.1** | 66.7 |
| Winogrande (XS) | 2 | 102 | **64.8** | 59.4 | **71.3** | 66.3 | **73.6** | 66.1 |

Table 4: Effect of shuffling answer options on the performance of Codex with MCP. Strong shuffle ensures the index associated with the correct answer choice changes. $N$ is the number of answer options for each dataset. $K$ exemplars are provided in the few-shot setting. Values marked with — could not be computed due to computational restraints (see Appendix B). Note that a slight drop in accuracy when shuffling is to be expected for many of the datasets where answer options refer to ordering (e.g., when an answer option is "both B and D are correct.")

# D  NON-CHERRY-PICKED MISSED QUESTIONS FROM COMMONSENSEQA

```
Question: James was looking for a good place to buy farmland.  Where
might he look?
A. midwest
B. countryside
C. estate
D. farming areas
E. illinois
Answer:

Question: What would vinyl be an odd thing to replace?
A. pants
B. record albums
C. record store
D. cheese
E. wallpaper
Answer:

Question: Aside from water and nourishment what does your dog need?
A. bone
B. charm
C. petted
D. lots of attention
E. walked
Answer:

Question: Though the thin film seemed fragile, for it's intended purpose
it was actually nearly what?
A. indestructible
B. durable
C. undestroyable
D. indestructible
E. unbreakable
Answer:

Question: What is someone who isn't clever, bright, or competent called?
A. clumsy
B. ineffectual
C. dull
D. clumsy
E. stupid
Answer:

Question: Blue read material outside of his comfort zone because he
wanted to gain what?
A. new perspective
B. entertained
C. understanding
D. hunger
E. tired eyes
Answer:

Question: What must someone do before they shop?
A. get money
B. have money
C. bring cash
D. go to market
E. bring cash
Answer:
```

Figure 23: Examples of CommonsenseQA questions missed by Codex with MCP (not cherry-picked). Answers are A, E, D, D, E, A, and A. Model selections were D, B, E, E, C, C, and B.

# E  TRADITIONAL PROMPT PERFORMANCE BY NORMALIZATION METHOD

| Dataset | N | K | Zero-Shot | | | One-Shot | | | Few-Shot | | |
|---|---|---|---|---|---|---|---|---|---|---|---|
| | | | Raw | LN | UN | Raw | LN | UN | Raw | LN | UN |
| AG News | 4 | 38 | **68.2** | 66.4 | 44.8 | 73.0 | **77.6** | 38.9 | **90.1** | 89.4 | 26.4 |
| ANLI R1 | 3 | 27 | 41.2 | **45.3** | 41.8 | **35.6** | 35.5 | 34.5 | 58.0 | **58.4** | 34.7 |
| ANLI R2 | 3 | 26 | 36.3 | 37.9 | **39.2** | **35.7** | 35.5 | 34.1 | 51.4 | **51.8** | 34.3 |
| ANLI R3 | 3 | 26 | 34.5 | **37.8** | 37.0 | 35.2 | **35.5** | 34.5 | **54.2** | 54.1 | 30.4 |
| ARC (Challenge) | 4 | 50 | 55.7 | 58.6 | **58.9** | 61.4 | 63.7 | **64.1** | 63.1 | **66.6** | 64.9 |
| ARC (Easy) | 4 | 57 | **84.2** | 82.9 | 78.0 | 85.8 | **85.9** | 81.2 | 87.1 | **87.8** | 82.2 |
| CODAH | 4 | 63 | 55.7 | **56.8** | 55.7 | 61.1 | **65.4** | 62.0 | 69.0 | **73.6** | 69.9 |
| CommonsenseQA | 5 | 79 | 65.4 | 57.9 | **68.5** | 70.2 | 70.9 | **73.1** | 77.1 | **78.6** | 77.6 |
| COPA | 2 | 113 | **92.0** | 86.0 | 90.0 | **95.0** | 92.0 | 92.0 | **96.0** | **96.0** | **96.0** |
| Cosmos QA | 4 | 24 | 32.6 | **43.0** | 34.6 | 35.9 | **44.0** | 36.4 | 30.1 | 31.3 | **38.1** |
| DREAM | 3 | 7 | **72.7** | 71.2 | 71.5 | 81.5 | **82.5** | 80.0 | 84.2 | **84.3** | 82.0 |
| Fig-QA | 2 | 99 | 73.2 | 74.0 | **79.6** | 76.8 | 79.5 | **82.4** | 79.0 | 81.8 | **82.5** |
| HellaSwag | 4 | 16 | — | — | — | — | — | — | — | — | — |
| LogiQA | 4 | 16 | 25.5 | 30.0 | **36.6** | 26.1 | 29.5 | **37.5** | 25.7 | 30.9 | **37.8** |
| MedMCQA | 4 | 58 | 34.1 | 37.6 | **37.8** | 37.8 | 41.1 | **42.1** | 38.0 | **41.2** | 41.2 |
| MMLU | 4 | 5 | 46.1 | 48.9 | **49.5** | — | — | — | — | — | — |
| OpenBookQA | 4 | 83 | 36.0 | 47.0 | **63.2** | 41.8 | 51.0 | **64.0** | 46.4 | 57.0 | **71.2** |
| PIQA | 2 | 35 | 82.7 | **83.7** | 68.6 | 83.2 | **84.1** | 67.4 | 84.5 | **86.1** | 70.7 |
| RACE-h | 4 | 4 | 48.5 | **52.3** | 49.3 | 49.5 | **53.2** | 52.3 | 51.3 | **55.2** | 53.0 |
| RACE-m | 4 | 8 | 64.2 | **67.5** | 63.3 | 67.3 | **70.5** | 66.0 | 68.9 | **71.7** | 67.5 |
| RiddleSense | 5 | 59 | **79.8** | 68.7 | 77.4 | **89.1** | 84.4 | 86.3 | **91.3** | 86.4 | 88.2 |
| Social IQa | 3 | 72 | 51.1 | 50.7 | **52.1** | 53.7 | **58.1** | 55.5 | 59.1 | **62.4** | 58.2 |
| StoryCloze | 2 | 44 | 76.3 | **80.3** | 79.5 | 80.1 | **83.4** | 82.4 | 84.7 | **88.2** | 86.9 |
| Winogrande (XL) | 2 | 102 | **62.5** | 62.4 | 60.1 | **71.6** | 70.8 | 66.3 | **75.5** | **75.5** | 68.7 |
| Winogrande (XS) | 2 | 102 | **63.0** | 62.7 | 59.8 | **71.0** | 69.9 | 64.6 | **76.2** | 75.8 | 69.4 |

Table 5: Effect of normalization strategy on the performance of Codex with cloze prompts. Raw is the strategy of selecting an answer choice based on raw probabilities. LN and UN are selecting an answer choice after normalizing probabilities based on length or unconditional completion probability respectively (see Brown et al. (2020)). N is the number of answer options for each dataset. K exemplars are provided in the few-shot setting. The best strategy for each dataset and exemplar count is bolded. Values marked with — could not be computed due to computational restraints (see Appendix B).

# F MEDMCQA TEST PERFORMANCE BY SUBJECT

| Subject | Accuracy |
|---|---|
| Anaesthesia | 52.5 |
| Anatomy | 47.5 |
| Biochemistry | 64.8 |
| Dental | 55.9 |
| ENT | 57.0 |
| FM | 56.8 |
| Medicine | 64.5 |
| Microbiology | 54.5 |
| O&G | 55.5 |
| Ophthalmology | 57.6 |
| PSM | 60.1 |
| Pathology | 65.2 |
| Pediatrics | 53.7 |
| Pharmacology | 64.4 |
| Physiology | 60.6 |
| Psychiatry | 33.3 |
| Radiology | 63.0 |
| Skin | 61.7 |
| Surgery | 52.9 |
| Unknown | 58.5 |

Table 6: Test accuracy of Codex with MCP on the MedMCQA test set by subject.

## G  MMLU PERFORMANCE BY TASK

| Subject | Zero-Shot | One-Shot | Five-Shot |
|---|---|---|---|
| abstract_algebra | 29.0 | 29.0 | **31.0** |
| anatomy | 62.2 | 59.3 | **65.9** |
| astronomy | 73.0 | 79.6 | **81.6** |
| business_ethics | 69.0 | **72.0** | 71.0 |
| clinical_knowledge | 70.2 | 70.9 | **71.7** |
| college_biology | 75.0 | 78.5 | **81.2** |
| college_chemistry | **50.0** | 45.0 | 42.0 |
| college_computer_science | 53.0 | 54.0 | **57.0** |
| college_mathematics | 35.0 | **39.0** | 37.0 |
| college_medicine | 65.3 | 71.7 | **72.3** |
| college_physics | 42.2 | **46.1** | 45.1 |
| computer_security | 75.0 | 75.0 | **79.0** |
| conceptual_physics | 63.8 | 65.1 | **66.0** |
| econometrics | 44.7 | 50.0 | **50.9** |
| electrical_engineering | 55.9 | 64.8 | **69.7** |
| elementary_mathematics | 48.9 | 53.4 | **54.8** |
| formal_logic | 27.8 | 49.2 | **54.0** |
| global_facts | 34.0 | **48.0** | 45.0 |
| high_school_biology | 80.0 | 81.3 | **85.2** |
| high_school_chemistry | 53.2 | **56.2** | 55.2 |
| high_school_computer_science | 79.0 | 77.0 | **80.0** |
| high_school_european_history | 80.0 | 85.5 | **86.7** |
| high_school_geography | 79.8 | **84.8** | 84.3 |
| high_school_government_and_politics | 89.1 | 92.2 | **93.8** |
| high_school_macroeconomics | 66.7 | **73.1** | 71.5 |
| high_school_mathematics | 37.4 | **40.7** | 40.4 |
| high_school_microeconomics | 69.3 | 70.6 | **74.4** |
| high_school_physics | 39.1 | **44.4** | 41.7 |
| high_school_psychology | 85.9 | 87.3 | **89.9** |
| high_school_statistics | 54.2 | 59.7 | **61.6** |
| high_school_us_history | 83.8 | 83.3 | **87.7** |
| high_school_world_history | 81.0 | 84.4 | **86.1** |
| human_aging | **74.0** | 77.1 | **74.0** |
| human_sexuality | 80.9 | **81.7** | 80.9 |
| international_law | 76.9 | 82.6 | **85.1** |
| jurisprudence | 80.6 | 81.5 | **85.2** |
| logical_fallacies | **81.0** | 74.2 | 79.1 |
| machine_learning | 45.5 | 50.0 | **54.5** |
| management | 74.8 | 83.5 | **86.4** |
| marketing | 61.5 | 87.2 | **89.7** |
| medical_genetics | 72.0 | 69.0 | **74.0** |
| miscellaneous | 85.8 | 85.4 | **87.7** |
| moral_disputes | 13.6 | 75.1 | **80.1** |
| moral_scenarios | 24.7 | **47.5** | 46.3 |
| nutrition | 72.2 | 72.2 | **76.5** |
| philosophy | 74.3 | 74.6 | **75.9** |
| prehistory | 74.1 | 78.7 | **81.2** |
| professional_accounting | **51.8** | 50.7 | 49.3 |
| professional_law | 52.2 | 53.9 | **54.8** |
| professional_medicine | 73.5 | **73.9** | 72.1 |
| professional_psychology | 69.9 | 74.2 | **74.5** |
| public_relations | 66.4 | 70.0 | **73.6** |
| security_studies | 67.3 | 73.5 | **75.5** |
| sociology | 80.6 | 86.6 | **87.6** |
| us_foreign_policy | **87.0** | 86.0 | **87.0** |
| virology | 50.0 | 51.8 | **54.8** |
| world_religions | 52.0 | 84.2 | **85.4** |

Table 7: Test accuracy of Codex with MCP on the MMLU test set by task.

