# OpenReview forum: "Leveraging Large Language Models for Multiple Choice Question Answering"
_ICLR.cc/2023/Conference — ICLR 2023 poster_

### Official Review · Reviewer_nm2P · 2022-10-24

**Confidence:** 4
**Correctness:** 4
**Technical Novelty And Significance:** 2
**Empirical Novelty And Significance:** 3
**Recommendation:** 8

**Clarity, Quality, Novelty And Reproducibility:**

The paper is quite well written and easy to follow and the code will be made available which facilitates predictability.
The paper presents an interesting and thorough empirical analysis but there is not much novelty from the methods/modeling side.

**Strength And Weaknesses:**

Strengths:
The paper presents an interesting idea for Multiple-Choice Question-Answering (using the answer symbol instead of the answer itself), motivates the idea well and does a thorough analysis over multiple datasets, and LLMs to analyze its performance in different settings (including few-shot settings).

Weaknesses:
There is no contribution/novelty from the modeling/methods side. The paper is mainly an empirical analysis of a different way of formulating an existing problem.

**Summary Of The Paper:**

The paper tackles the problem of using Large Pre-trained Language Models (LLMs) for multiple-choice question-answering. Instead of using the standard cloze formulation, the paper suggests presenting the question and answer choices to the model and have the model output the answer symbol (e.g., A, B,C, ...etc.). The authors conduct a thorough study on multiple datasets and using different LLMs and show that the new formulation improves over variants of the standard cloze prompting technique. In addition, they measure the effect of shuffling the order of answers on the performance and note which LLMs show better invariance to the order of the answers.

**Summary Of The Review:**

Overall, while the paper doesn't have modeling/methods novelty, it presents a thorough and interesting analysis of how a different formulation of Multiple-Choice Question-Answering affects the performance of LLMs on the task. The analysis includes multiple datasets and several LLMs and looked at issues affecting the performance such as order invariance which are sometimes ignored in other studies.

---

> ### Author Response · Authors · 2022-11-16
> **Author Response**
>
> Thank you for the encouraging review! With respect to your comments about the novelty of the paper, please see our response in the shared response to all reviewers.

---

> > ### Comment · Reviewer_nm2P · 2022-12-08
> > **Thanks to the authors for the detailed response. While I still don't see technical novelty as a strength of this paper, I see that it is a thorough paper and the empirical analysis itself has some novelty and is insightful.**
> >
> > Thanks to the authors for the detailed response. While I still don't see technical novelty as a strength of this paper, I see that it is a thorough paper and the empirical analysis itself has some novelty and is insightful.

---

### Official Review · Reviewer_jqnn · 2022-10-24

**Confidence:** 4
**Correctness:** 4
**Technical Novelty And Significance:** 2
**Empirical Novelty And Significance:** 3
**Recommendation:** 5

**Clarity, Quality, Novelty And Reproducibility:**

The paper is well written and I believe the experiments are verifiable with the given information, i.e. it should be possible to reproduce them.

Regarding novelty, I am less convinced. The authors mention others having used the MCP approach. So the main addition here is the systematic discussion and wide range of experiments.


**Strength And Weaknesses:**

Strengths: The authors explain their approach well. They also discuss the (somewhat surprising) variance between different models in their ability to separate the letter from the answer. (They call this Multiple Choice Symbol Binding.) The approach is evaluated on a wide range of (20) datasets.

Weaknesses: The approach is not new, just discussed and evaluated. The authors differentiate their suggested prompting from “prompt engineering”, which they seem to define as fine-tuning of prompts to increase model performance. However, I’m not convinced that these are fundamentally different, and would include research such as theirs in the general domain of prompt engineering.

**Summary Of The Paper:**

The authors identify a better method to prompt LLMs for multiple-choice question answering. Instead of (the usual) comparing the probability of producing each answer, they present all options to the model and then identify the right option by producing just the letter that identifies the answer.


**Summary Of The Review:**

The authors discuss an alternative (but not novel) way to prompt LLMs for better results on multiple-choice tasks. The prompt is well-motivated and thoroughly discussed with a good range of experiments that support the author's arguments. However, it is not novel: it is a fairly obvious way to prompt and has been tried before.

---

> ### Author Response · Authors · 2022-11-16
> **Author Response**
>
> Thank you for your insights and helpful feedback. We address your concerns below:
>
> > The approach is not new, just discussed and evaluated.
> > Regarding novelty, I am less convinced. The authors mention others having used the MCP approach. So the main addition here is the systematic discussion and wide range of experiments.
> > It is a fairly obvious way to prompt and has been tried before.
>
> Please see our response to this concern in the shared response to all reviewers.
>
> > The authors differentiate their suggested prompting from “prompt engineering”, which they seem to define as fine-tuning of prompts to increase model performance. However, I’m not convinced that these are fundamentally different, and would include research such as theirs in the general domain of prompt engineering.
>
> Thanks for pointing this out. We agree that this research falls in the general domain of prompt engineering and have made changes to the paper to reflect this. In particular, we’ve renamed the “Prompt Engineering” section to “Prompt Phrasing” and reworded it. References to prompt engineering in the Introduction and Conclusion sections have also been changed.

---

### Official Review · Reviewer_rg4x · 2022-10-25

**Confidence:** 4
**Correctness:** 3
**Technical Novelty And Significance:** 2
**Empirical Novelty And Significance:** 2
**Recommendation:** 5

**Clarity, Quality, Novelty And Reproducibility:**

Clarity
Good. But it would be better to include more analysis.

Quality
Good. A simple but effective prompting method.

Novelty
Novelty is limited. Details can be found in the weakness part.

Reproducibility
Good. The code is attached by the authors.


**Strength And Weaknesses:**

Strength:

1. This paper is well-written and easy to understand.

2. The authors propose a simple but effective prompting method, which outperforms previous CP methods, approaching or even surpassing SOTA performance.

3. Experimental results show that the MCQA ability of LLMs has been previously underestimated. And there is a better way to prompt a single LLM. The potential of multiple choices prompts can be further tapped. Future work include prompt engineering is still promising.

Weakness:

1. The novelty of this paper is limited. Multiple choices prompting (MCP) has been used in other QA tasks, such as TruthfulQA and RACE.

2. Although the experimental results prove that the proposed multiple choices prompting (MCP) methods can outperform existing cloze prompting (CP) methods, the reasons behind it are still unclear. Since the authors have listed several problems within CP methods, I'm curious about whether these problems are all solved or avoided by their MCP methods. More analysis is needed to show this.


**Summary Of The Paper:**

The authors find that the MCQA ability of LLMs has been underestimated, and they propose multiple choice prompting (MCP), in which a question and its symbol-enumerated candidate answers are all passed to an LLM as a single prompt. Surprisingly, the performance of LLMs equipped with MCP dramatically improves, approaching or even surpassing SOTA. This demonstrates that the power of LLMs can be used broadly in the future.

**Summary Of The Review:**

Interesting paper but not good enough. It would be better to include more analysis on whether MCP can deal with these several problems that CP faces.

---

> ### Author Response · Authors · 2022-11-16
> **Author Response**
>
> Thank you for the thoughtful review and valuable recommendations. We address your concerns below:
>
> > The novelty of this paper is limited. Multiple choices prompting (MCP) has been used in other QA tasks, such as TruthfulQA and RACE.
>
> Please see our response to this concern in the shared response to all reviewers.
>
> > Although the experimental results prove that the proposed multiple choices prompting (MCP) methods can outperform existing cloze prompting (CP) methods, the reasons behind it are still unclear. Since the authors have listed several problems within CP methods, I'm curious about whether these problems are all solved or avoided by their MCP methods. More analysis is needed to show this.
>
> Thanks for the recommendation. We agree with the assessment and have substantively expanded the paper to address it. Please see the Results section for the majority of the relevant changes. In the revised version of this section we discuss how MCP avoids the issues outlined in Section 3. MCP is less expensive (we now quantify this by pointing out that on average the MCP columns in Table 2 required 4.3x less API calls (or equivalently forward passes with batch size 1) than did the CP columns). MCP clearly doesn’t require any normalization strategies for good performance (as supported by Table 2). To address the other two points from Section 3 (“No direct comparison between answers” and “Conflation of likelihood as answer and likelihood as natural language”) we evaluate accuracy for a diverse set of 3 datasets under different corruptions of the answer options. We show that answer choice corruptions like randomly adding spaces or changing casing result in 10+% accuracy loss for SCP methods but less than 2% loss for MCP.
>
> > Clarity Good. But it would be better to include more analysis.
>
> We assume this is in reference to the same concern addressed above. Please let us know if this is in reference to something else that could be improved.

---

> > ### Comment · Reviewer_rg4x · 2022-11-28
> > **Thanks**
> >
> > Thanks for the author's detailed responses, however, the novelty is still my big concern.

---

### Official Review · Reviewer_oxYV · 2022-10-25

**Confidence:** 4
**Clarity, Quality, Novelty And Reproducibility:** The submission is an analysis paper r…
**Correctness:** 2
**Technical Novelty And Significance:** 3
**Empirical Novelty And Significance:** 3
**Recommendation:** 5

**Strength And Weaknesses:**

(Strengths)
1) Reveals problematic ingredients for likelihood-based answering.
2) Introduce the concept of MCSB and measure it by PPA.
3) Concentrated results that significantly improves QA performance by using multiple-choice prompting.


(Weaknesses)
1) Individual problematic ingredients are neither being theoretically-proven nor empirically-proven.
2) No novel/brand new ideas. Mostly empirical analysis based on OpenAI playground.
3) Some major arguments are less supported.


**Summary Of The Paper:**

This paper addresses the difference between multiple-choice prompting and standard prompting (so called cloze prompting), clarifying major reasons why LLM underperforms on Multiple Choice Question Answering (MCQA) problems. First, what LLM tries to predict in terms of “more likely” does not always mean “more correctly”. This conflation often happens when the tokens in the answer sequence is less common or less grammatical. Second, LLM must rely on normalization schemes to compare candidate answers with different lengths or different frequencies. But this yields additional dependency on tokenizer. Third, standard prompting compares different options only indirectly via the (normalized) likelihood without direct comparison. Obviously, such standard prompting is expensive comparing to generating one option token.

To make LLM solve MCQA problems with order-invariance, the authors propose Multiple Choice Symbol Binding (MCSB) capability that could be model-agnostically testable by recording the answer with the highest probability for each ordering of question (so called PPA). The experimental results show that training on code data (especially by multi-staging) is useful for MCSB. Providing more shots as few-shot examples also help boosting the performance.


**Summary Of The Review:**

(Major concerns)
How to make sure Codex model clearly outperforms Instruct model? This is a critical question as the authors measures the main experiments (Table 2) that compare Multiple Chocie Prompting (MCP) and Cloze Prompting (CP) only with Codex model.

0) The capability to perform MCSB could be due to human feedback alignment by Reinforcement Learning rather than other points indicated by the authors.

1) Are the PPA difference between Codex and Instruct (in Figure 2) statistically significant? While no statistical test has been provided, it seems not easy to decline null hypothesis that says the difference is a random effect.

2) Only Codex tested on OpenBookQA shows strong performance gain when using MCP, whereas Instruct outperforms Codex on the other two tasks in Table 1. More detailed experiments are necessary to convince how Codex achieve such higher accuracy.


(Minor concerns)
1) Any reason to choose OpenBookQA which also matters the performance of retriever?

2) Do you know how Codex model is exactly trained? Codex model that you used could be first based on Instruct, then being further trained on code data. Equally likely, Codex model might perform it's own alignment similar to Instruct but based on the preference of generated codes.

---

> ### Author Response · Authors · 2022-11-16
> **Author Response**
>
> Thank you for your time and for your valuable feedback on how the paper might be improved. We address your concerns below:
>
> > Individual problematic ingredients are neither being theoretically-proven nor empirically-proven.
>
> We assume this is in reference to the same concerns you outline under the “Summary of the Review” section (which we address below). Please let us know if this is in reference to something else that could be improved.
>
> > No novel/brand new ideas.
>
> Please see our response to this concern in the shared response to all reviewers.
>
> > Some major arguments are less supported.
>
> We assume this is in reference to the same concerns you outline under the “Summary of the Review” section (which we address below). Please let us know if this is in reference to something else that could be improved.
>
> > How to make sure Codex model clearly outperforms Instruct model? This is a critical question as the authors measures the main experiments (Table 2) that compare Multiple Chocie Prompting (MCP) and Cloze Prompting (CP) only with Codex model. (0) The capability to perform MCSB could be due to human feedback alignment by Reinforcement Learning rather than other points indicated by the authors. (1) Are the PPA difference between Codex and Instruct (in Figure 2) statistically significant? While no statistical test has been provided, it seems not easy to decline null hypothesis that says the difference is a random effect. (2) Only Codex tested on OpenBookQA shows strong performance gain when using MCP, whereas Instruct outperforms Codex on the other two tasks in Table 1. More detailed experiments are necessary to convince how Codex achieve such higher accuracy.
>
> Thanks for bringing this to our attention. It was not our intention in the paper to claim that Codex is empirically stronger than Instruct, but we failed to convey that in our writing. In our revised version of the paper we state clearly in Section 5.1, “It is evident that both Codex and Instruct have high multiple choice symbol binding ability and can effectively leverage MCP prompts across tasks. We make no argument that one is empirically stronger than the other.” We then outline our reasons for choosing Codex: “For all our further experiments we choose to use Codex (Davinci) because it is the least expensive (since it is currently in free beta), and because Codex should not add any dataset leakage issues that were not already present in GPT-3 since it was exclusively fine-tuned on Python files.” To expand a little more on why cost was an important factor consider the AG News dataset, which has a test set of size 7,600. To calculate accuracy in the few-shot MCP setting with Instruct would cost COST_PER_TOKEN x NUM_INSTANCES x TOKENS_PER_INSTANCE = (0.02/1000) x 7600 x ~4,000 = ~608 dollars. In the non-MCP setting it would cost that multiplied by 8 (4 answer choices and 2 forward passes for each). So just for the few-shot setting of AG News we would need to pay ~$5,000-5,500. And that is for one setting of one dataset. That Codex is currently free made presenting a large array of results possible.
>
> > Any reason to choose OpenBookQA which also matters the performance of retriever?
>
> This is a great question, and we have made clarifying changes to the paper to address it (see “We evaluate on the OpenBookQA…available to the models.” in Section 4). It is true that the original OpenBookQA offers a “book” component (a list of elementary science facts from the WorldTree corpus), but we follow the practice from GPT-3 and later language model papers of ignoring the retrieval aspect of OpenBookQA. We choose OpenBookQA because it is widely used as a benchmark in large language model papers and because it is an archetypal multiple choice dataset.
>
> > Do you know how Codex model is exactly trained? Codex model that you used could be first based on Instruct, then being further trained on code data. Equally likely, Codex model might perform it's own alignment similar to Instruct but based on the preference of generated codes.
>
> We agree that exactly how the Codex model was trained was left somewhat ambiguous in the paper in which it was presented. Luckily, we were able to get an answer from the authors. Codex was trained from the original (non-Instruct) GPT-3. We include this information in the paper in Section 4. We also express in that section that further training (whether it be through human feedback alignment or through training on code) seems essential for good MCSB ability because Codex and Instruct both outperform GPT-3, which they were fine-tuned from, in terms of MCSB ability.

---

### Author Response · Authors · 2022-11-16
**Shared Response to All Reviewers**

We thank all reviewers for the substantive and thoughtful reviews.

Based on the helpful feedback we received we have made a number of changes to the paper and posted a revised version. These changes include better justifying our model and dataset choices, correcting the way we talk about our method with respect to prompt engineering, and adding additional quantitative analysis and an experiment with answer choice corruption to the Results section in order to demonstrate how MCP avoids the issues faced by SC that we enumerate in Section 3. We appreciate the feedback, which has helped us to meaningfully improve the paper.

A common reviewer concern was that our method is not sufficiently novel. This concern is understandable, particularly as we include references to other works that use MCP in an effort to make our cited work as complete as possible. However, though the MCP formulation has been used before, we believe that our work brings enough novel and important contributions to justify its publication at ICLR 2023. Our reasons are below:

First, existing work with MCP and large language models is limited, and we believe our work could meaningfully increase community adoption. As we state in our paper, the only LLM papers that use MCP are Gopher and followup Chinchilla. That is, MCP is not used for evaluation in many LLM papers (e.g., GPT-3, GLAM, Megatron-LM, PaLM, etc.). In the PaLM paper, which was published after Gopher and Chinchilla, the authors specifically mention that Gopher and Chinchilla use a different setup for RACE-h/m and thus don’t include the results (see Table 4 caption). Given that in our work we show that MCP greatly outperforms CP across most of 20 diverse datasets and that Codex+MCP achieves a new SOTA on 9 datasets, we believe future LLM paper authors will be more inclined to use MCP instead of sticking with the GPT-3 style prompts which systematically underestimate the performance of LLMs with high MCSB performance on MCQA tasks. To the best of our knowledge the only real prior discussion of MCP is found in the appendix of the Gopher paper and it is treated peripherally. Given the ~10% average accuracy gain of CP to MCP that we show across 20 datasets, we believe MCP deserves more than a peripheral treatment.

Second, our investigation on the importance of MCP may provide researchers with better explanations of observed phenomena. For example, in the Gopher paper the authors point out that Gopher outperforms GPT-3 by a notable 17%. They use MCP for Gopher for this task. The authors note, “Smaller models from the Gopher family do not perform as well on these tasks, which suggests that data alone does not explain the performance difference — the combination of scale and data is crucial.” It seems very possible that use of MCP is responsible for both some of the gain in accuracy and for the lower performance of smaller models (due to no emergence of MCSB ability).

Finally, many aspects of our paper are, to our knowledge, completely novel. A few of these are:
* We identify challenges faced by the CP approach - conflation of likelihood as answer and likelihood as natural language, reliance on normalization procedures, no direct comparison between answers, and computational expense. To our knowledge, previous work has cited none of these as reasons for using MCP.
* We show that MCP avoids the above issues, including showing that using MCP requires less forward passes/API calls, and that MCP is robust to simple corruptions of answer choices (like changing character caps or adding spaces) where CP is not.
* We show that normalization procedures are not necessary to get high performance on MCQA tasks when using LLMs.
* For the first time, we introduce the concept of MCSB ability and systematically demonstrate that different models have different degrees of this ability (we agree with Reviewer jqnn that this result is “somewhat surprising” and believe other researchers will also find it to be a surprising and useful result).
* To show the variance in MCSB ability between models we introduce a novel way of measuring the ability - PPA. We hope it will be useful to researchers in the future who would like to measure the MCSB ability of their models.
* For the first time, we identify failure to use multiple choice prompting as a reason for why large language models often lag behind the state of the art on multiple choice question answering tasks. We systematically support this novel attribution with extensive experiments across 20 diverse datasets.

Thanks again for your reviews and consideration.

---

### Decision · Program_Chairs · 2023-01-20

**Decision:**

Accept: poster

**Justification For Why Not Higher Score:**

Limited novelty and scope.


**Justification For Why Not Lower Score:**

Clarity, exhaustive experiments, simplicity/practicality of the method.


**Metareview: Summary, Strengths And Weaknesses:**

Summary: The authors study how the performance of language models on multiple choice questions is under-estimated by current prompts and propose a solution. Their solution relies on a new prompt which enumerates the multiple choices and ask the model to choose one by referring to its numbering. This allows the model to focus on answer correctness rather than favoring answers with more common formulations.

Strengths:
- simple and effective method, easy to apply and reproduce.
- clarity of the paper, reference to related work.
- empirical results are convincing.

Weaknesses:
- technical and empirical novelty of the paper is limited (all reviewers rated 2-3 on these aspects).
- the paper has a limited scope as it excludes more exhaustive prompt engineering (see Section 5.4).


**Note From Pc:**

if the above contains the word "oral" or "spotlight" please see: "oral" presentation means -> notable-top-5% and "spotlight" means -> notable-top-25%. As stated in our emails, we are disassociating presentation type from AC recommendations